# Intraspecific Variability in the Toxin Production and Toxin Profiles of In Vitro Cultures of *Gambierdiscus polynesiensis* (Dinophyceae) from French Polynesia

**DOI:** 10.3390/toxins11120735

**Published:** 2019-12-17

**Authors:** Sébastien Longo, Manoella Sibat, Jérôme Viallon, Hélène Taiana Darius, Philipp Hess, Mireille Chinain

**Affiliations:** 1Laboratoire de recherche sur les Biotoxines Marines Institut Louis Malardé-UMR 241 EIO, 98713 Papeete-Tahiti, French Polynesia; jviallon@ilm.pf (J.V.); tdarius@ilm.pf (H.T.D.); mchinain@ilm.pf (M.C.); 2Laboratoire Phycotoxines, IFREMER, Rue de l’Ile d’Yeu, 44311 Nantes, France; Manoella.Sibat@ifremer.fr (M.S.); Philipp.Hess@ifremer.fr (P.H.)

**Keywords:** *Gambierdiscus polynesiensis*, ciguatera, ciguatoxins, gambierone, 44-methylgambierone, LC-MS/MS, CBA-N2a, toxin profiles

## Abstract

Ciguatera poisoning (CP) is a foodborne disease caused by the consumption of seafood contaminated with ciguatoxins (CTXs) produced by dinoflagellates in the genera *Gambierdiscus* and *Fukuyoa*. The toxin production and toxin profiles were explored in four clones of *G. polynesiensis* originating from different islands in French Polynesia with contrasted CP risk: RIK7 (Mangareva, Gambier), NHA4 (Nuku Hiva, Marquesas), RAI-1 (Raivavae, Australes), and RG92 (Rangiroa, Tuamotu). Productions of CTXs, maitotoxins (MTXs), and gambierone group analogs were examined at exponential and stationary growth phases using the neuroblastoma cell-based assay and liquid chromatography–tandem mass spectrometry. While none of the strains was found to produce known MTX compounds, all strains showed high overall P-CTX production ranging from 1.1 ± 0.1 to 4.6 ± 0.7 pg cell^−1^. In total, nine P-CTX analogs were detected, depending on strain and growth phase. The production of gambierone, as well as 44-methylgamberione, was also confirmed in *G. polynesiensis*. This study highlighted: (i) intraspecific variations in toxin production and profiles between clones from distinct geographic origins and (ii) the noticeable increase in toxin production of both CTXs, in particular CTX4A/B, and gambierone group analogs from the exponential to the stationary phase.

## 1. Introduction

Ciguatera poisoning (CP) is a common seafood intoxication mainly found in tropical to subtropical coral reef areas. CP cases occur after the consumption of seafood contaminated with toxins known as ciguatoxins (CTXs) that are produced by dinoflagellates in the genera *Gambierdiscus* and *Fukuyoa*. CTXs enter the coral reef food web through grazing by herbivores and detritivores and are further accumulated through predation and biotransformed in carnivores [1,2]. 

CP is characterized by complex symptomatology, including gastrointestinal, cardiovascular, and neurological signs, which may persist in some form for weeks, months, and even years [3,4]. This symptomatology may vary in different parts of the world: e.g., neurological signs clearly predominate in the Pacific region, whereas gastrointestinal symptoms seem to prevail in the Caribbean and the Indian Ocean, with the report of additional mental status alterations in this latter region [5,6,7]. These regional differences in symptom patterns have been attributed to the presence of different suites of CTXs in different geographic regions [6,8], the occurrence of different *Gambierdiscus/Fukuyoa* species/strains depending on the region [9], or the presence of other co-occurring toxins in contaminated marine products [10]. Highest incidence rates of CP are consistently reported from the Pacific Island Countries and Territories (PICTs) where it constitutes not only a serious threat for human health but also has major socioeconomic impacts on the life of local populations [11,12,13,14].

Overall, 18 species of Gambierdiscus are now recognized worldwide: *G. toxicus*, *G. belizeanus*, *G. australes*, *G. pacificus*, *G. polynesiensis*, *G. caribaeus*, *G. carolinianus*, *G. carpenteri*, *G. excentricus*, *G. scabrosus*, *G. silvae*, *G. balechii*, *G. cheloniae*, *G. lapillus*, *G. honu*, *G. jejuensis*, *G. lewesii*, and *G. holmesii* [15,16,17,18,19,20,21]. In addition, two globular *Gambierdiscus* species have recently been reclassified as *Fukuyoa yasumotoi* and *F. ruetzleri*, with a new species described as *F. paulensis* [18,22,23]. These organisms are the potential source of several groups of bioactive compounds, including CTXs [24,25,26], maitotoxins (MTXs) [27,28,29], gambieric acids [30], gambierol [31,32], gambieroxide [33], and gambierones [34,35,36]. While CTXs and, to a lesser extent, MTXs have been implicated in CP [37,38], it is not fully understood if the other metabolites could also play a role in CP, but most of them are considered as compounds of interest for their bioactivity [38] and/or potential therapeutic applications [34,39]. 

Several *Gambierdiscus* and *Fukuyoa* species/strains have been identified as likely producers of CTXs or CTX-like compounds, as measured by Mouse Biological Assay (MBA) [15,21,40,41], Receptor Binding Assay (RBA) [24], neuroblastoma cell-based assay (CBA-N2a) [42,43,44,45], fluorescent calcium flux assay [46], or liquid chromatography–tandem mass spectrometry (LC-MS/MS) [47,48,49,50,51]. However, the formal confirmation of the presence of previously characterized CTX- and/or MTX-group compounds using LC-MS/MS has been demonstrated only in a limited number of species, i.e., *F. paulensis*, *G. australes*, *G. caribaeus*, *G. excentricus*, *G. pacificus*, *G. polynesiensis*, and *G. toxicus*, although for this latter species, a potential misidentification of the species is likely [38]. Conversely, most species of *Gambierdiscus* seem to produce 44-methylgambierone (previously reported as MTX3) [20,21,49,52,53]. Most of these toxigenic species/strains, however, exhibit very low toxicity, toxin production ranging from several femtograms to sub-picograms per cell [38,54]. Only two species proved significantly more toxic than the others, producing pg amounts of CTXs, namely *G. polynesiensis* [24,49,51] and *G. excentricus* [44,54,55], and are recognized as important toxin-producing species in the South Pacific and the Atlantic Oceans, respectively. As an example, published toxicity records for *G. polynesiensis* range from 1.2 up to 18.2 pg cell^−1^, as determined by either RBA [24], CBA-N2a [47,56], or LC-MS/MS [49].

The overall toxicity of *Gambierdiscus* species/strains is believed to vary according to the types of CTX congeners they produce, in addition to the amount [56,57]. Based on their skeletal structures, CTX analogs can be separated into two types, i.e., CTX1B type and CTX3C type analogs [57]. Once accumulated in fish, algal CTXs are metabolized into more oxidized congeners [2,58,59], although the occurrence of small amounts of the oxidized forms of CTXs, i.e., 2,3-dihydroxy-P-CTX3C, 51-hydroxy-P-CTX3C, 2-hydroxy-P-CTX3C, M-seco-P-CTX3C has also been shown in cultures of *G. toxicus* and *G. polynesiensis* [26,51]. Complex biotransformation processes likely take place in fish livers [2] and lead to species-specific and region-specific toxin profiles in fish [26,57]. More precisely, Ikehara et al. (2017) [2] have recently showed that the enzymatic oxidation of CTX4A, CTX4B, and CTX3C, which are produced by the algal source-organism, actually leads to CTX derivatives that bioaccumulate in fish, namely CTX-1 (or CTX1B), CTX-2 (or 52-epi-54-deoxyCTX1B), CTX-3 (or 54-deoxyCTX1B), 2-hydroxyCTX3C, and 2,3-dihydroxyCTX3C. Such findings, together with results of studies that compared the toxin characteristics in various fish species vs. the causative *Gambierdiscus* spp. from different areas in Japan, strongly suggest that the genetic composition of *Gambierdiscus* populations in toxic areas likely contributes to shape the toxin profiles in fish [57,60]. Hence, extensive studies aiming at documenting the toxicity level and complete toxins profiles in *Gambierdiscus/Fukuyoa* spp. worldwide are most useful to understand/predict CP patterns in ciguatera-prone areas.

Numerous studies have examined the responses of *Gambierdiscus* spp. (i.e., growth and toxin production) to varying environmental factors including temperature, salinity, light, and nutrient availability [38,60,61,62,63,64]. Results point out the complexity of the toxinogenesis in *Gambierdiscus/Fukuyoa* spp. and suggest that this functional trait may be under the control of a combination of factors, including genetic [15,54,65,66], physiological [24,65,67], environmental [43,68] and microbial drivers [69]. These variations in *Gambierdiscus/Fukuyoa* species/strains affect not only the nominal cell toxin content but may also concern the toxin profiles, i.e., both the identity and ratio of the different CTXs congeners produced [24,50]. 

In French Polynesia, *G. polynesiensis* is regarded as the dominant source of CTXs entering the food web [56]. Thus, CP risk assessment programs currently conducted in this region primarily focus on the monitoring of the abundance of this key toxin-producing species using species-specific molecular assays [54,56]. At least 11 distinct P-CTX congeners have been previously characterized in this species, namely P-CTX4B, P-CTX4A (52-epi-P-CTX4B), P-CTX3C, P-CTX3B (49-epi-P-CTX3C), M-seco-P-CTX3C methyl acetal, 2-hydroxy-P-CTX3C, 54-deoxy-P-CTX1B, 52-epi-54-deoxy-P-CTX1B, 51-hydroxy-P-CTX3C, M-seco-P-CTX3C, and M-seco-P-CTX4A/4B [24,47,70,71]. 

Several studies have highlighted the significant variations observed in CTX production across species/strains from different geographic origins [15,49,54,65,66]. The objective of the present study was to characterize the toxin production and profiles in four clones of *G. polynesiensis* from distinct geographic origins, at two distinct growth phases, i.e., exponential vs. stationary growth phase, focusing primarily on three groups of toxins, i.e., CTXs, MTXs, and gambierone group analogs. For this purpose, CTXs were detected using both CBA-N2a and LC-MS/MS whereas only this latter method was performed for the detection of MTXs and gambierone group analogs. Furthermore, CTX-related toxicity, as determined by CBA-N2a, and concentrations of CTX-analogs, as determined by LC-MS/MS analysis, will also be compared to each other to evaluate whether or not there is an agreement between these two methods.

## 2. Results

### 2.1. Growth Rates 

Cultivated in batch culture conditions (26 ± 1 °C, stabilized pH at 8.4, under 60 ± 10 μmol photons m^−2^ s^−1^ irradiance, in a 12 h light: 12 h dark photoperiod), all clones displayed a typical growth curve characterized by an exponential phase, from day 0 to 12, and a stationary phase between day 12 to 18 (Figure 1).

Growth rates monitored at the exponential phase ranged between 0.22 ± 0.01 and 0.35 ± 0.09 div. day^−1^ (Table 1), with no significant differences between clones (*p*-values ranging from 0.1 to 1, Wilcoxon test). The total cell yields obtained at the stationary phase ranged from 275,000 to 440,000 cells (in a total volume of 180 mL in 250 mL-Erlenmeyer flasks) as assessed by automated counting on the Multisizer III^TM^ particle counter (Beckman Coulter Inc., Brea, CA, USA), with clone NHA4 giving the highest cellular biomass. No cell size differences have been noticed.

### 2.2. CBA-N2a Toxicity Data

All four *G. polynesiensis* clones in this study produced CTX activity, as measured by the CBA-N2a assay (Figure 2). Indeed, all dichloromethane-soluble fractions (DSF) showed no cytotoxicity on neuroblastoma cells in OV^−^ conditions (data not shown), whereas a sigmoidal dose–response curve was observed in OV^+^ conditions, a response typical of CTX activity.

The effective concentration reducing 50% of cell viability (EC_50)_ and the overall composite toxicity were estimated at both growth phases for each *G. polynesiensis* clone showing no significant difference between exponential vs. stationary growth phase regardless of the clone (Table 2). Among these four clones, NHA4 produced the highest toxin content (3.7 ± 0.3 pg P-CTX3C eq. cell^−1^) at the late exponential phase, followed by RIK7, RAI-1, and RG92, this latter being significantly less toxic than the other three strains (Table 2).

### 2.3. LC-MS/MS Analyses and Toxin Profiles in G. polynesiensis

#### 2.3.1. Quantification of CTX Analogs

LC-MS/MS analyses of the crude methanol extracts of the four clones revealed the presence of nine detectable P-CTX congeners (Figure 3 and Figure 4, Table 3) consisting of (i) two majors compounds, P-CTX3B and P-CTX3C, and (ii) seven minor CTXs analogs: two nonpolar congeners from the P-CTX1B group, i.e., P-CTX4A and P-CTX4B; two oxidized forms of P-CTX3C, i.e., 2-OH-P-CTX3C and M-seco-P-CTX3C; and three hitherto undescribed isomers belonging to the P-CTX3B/C group, referred to as P-CTX3B/C group isomers 1, 2, and 3 (Figure 3 and Table 3). 

Of note, only P-CTX3B/C group isomers 2 and 3 were present at quantifiable amounts, whereas P-CTX3B/C isomer 1 (RT = 7.28 min) was consistently detected at levels below the limit of quantification (LOQ). 

When detected above the LOQ, the quantitation of each other P-CTX congener was estimated by integration of the chromatogram peaks (Figure 3) allowing for the calculation of the overall toxin cell content in each clone (Table 3). P-CTX3B and -3C amounts accounted for 54%–69% and 26%–32%, respectively, of the total CTX production. Interestingly, if the amount per cell of P-CTX3B remained stable or slightly increased from the exponential to the stationary growth phase, its relative abundance clearly decreased in favor of minor analogs at the stationary phase. Still, P-CTX3B and P-CTX3C contributed to at least 80% of the total CTXs profile in the stationary phase. Regarding the total toxin content, NHA4 was the highest producer, i.e., 4.61 ± 0.71 pg P-CTX3C eq. cell^−1^, followed by RIK7, RA-1, and RG92, this latter being again the least toxic strain (Table 3). Thus, the same hierarchy was found among these four strains whatever the growth phase between LC-MS/MS and CBA-N2a results.

The relative abundance of P-CTX3B/C group isomers 2 and 3 differed, depending on the strain and growth stage considered: for instance, in strain RIK7 examined at the stationary phase, these compounds represented up to 10% of the total CTX production. In addition, P-CTX3B/C group isomer 3 always contributed to 5% of the total CTXs production in all four clones at the stationary phase (Figure 4).

Analysis of toxin profiles also revealed a noticeable increase in the diversity of CTX analogs from the exponential to the stationary phase (Table 3 and Figure 4). Strains RIK7 and NHA4 showed the most diversified toxin profiles, which consisted of the two primary toxins P-CTX3B and P-CTX3C at the exponential phase vs. seven analogs at the stationary phase (Figure 4, Table 3). 

Indeed, the total CTX content found in RG92 was significantly lower compared to RIK7, NHA4, and RAI-1 (Table 4). Conversely to CBA-N2a results, the ANOVA analysis reveals significant differences in toxin production estimated by LC/MS/MS between the exponential vs. stationary phase (*p* < 0.05; i.e., 0.00032), with a 24% increase in toxicity in average (i.e., 0.7 pg cell^−1^) at the stationary phase (Table 4). ANOVA on the differences in toxicity between clones and growth phases indicates that clone RG92 was significantly less toxic than RAI-1, RIK7, and NHA4, and that toxin production of CTXs significantly increased from the exponential to the stationary growth phase (Table 4).

Finally, CTX contents estimated between CBA-N2a (Table 2) and LC-MS/MS (Table 3) in *G. polynesiensis* clones were also compared to evaluate the good agreement between these two detection methods. At the exponential growth phase, CBA-N2a data were in good agreement with those of LC-MS/MS. However, as a significant increase in toxin production with the age of the culture was observed only by LC-MS/MS (ANOVA analysis, *p* = 0.0003; Table 4), there is somewhat more discrepancy between the CBA-N2a results and total concentrations by LC-MS/MS. However, toxicity equivalency factors (TEFs) of all analogs detected by LC-MS/MS are not known to date and thus the differences in total contents (typically in an order of 25% magnitude) should not be over-interpreted.

#### 2.3.2. Quantification of the Gambierone Group Compounds

Cell and culture media extracts of strains RIK7, NHA4, RAI-1, and RG92 were also tested for the presence of compounds classically found in *Gambierdiscus* spp. cultures, i.e., the gambierone group analogs. All four *G. polynesiensis* strains were found to produce gambierone and 44-methylgambierone, e.g., strain NHA4 (Figure 5).

Table 5 gives the relative abundance of these two compounds in cells (intracellular) vs. in culture media (extracellular), at the exponential vs. stationary phase. Regardless of the growth phase considered, gambierone was consistently produced in higher amounts (2- to 8-fold) than 44-methylgambierone with concentrations ranging from 46.1 to 229.8 and 5.8 to 74.1 pg MTX1 eq. cell^−1^, respectively, in *G. polynesiensis* cells. Additionally, in the culture media of *G. polynesiensis* clones, gambierone and 44-methylgambierone concentrations ranged from 1.3 to 121.3 and <LOQ to 88.1 pg MTX1 eq. cell^−1^, respectively. Similar to intracellular CTXs detected by LC-MS/MS, the production pattern seems to reveal an enhanced biosynthesis of these metabolites from the exponential to the stationary growth phase. Of note, NHA4, which yielded the highest cellular biomass at the stationary phase (data not showed), presented the lowest amounts of gambierone. Strain NHA4 was also the only strain in which gambierone and 44-methylgambierone concentrations were higher in the culture media (67% and 84%, respectively) than in the cells (33% and 16%, respectively) at stationary growth phase. Since only one batch culture was analyzed per clonal isolate (in a single run), it was not possible to assess whether the toxicity variations observed among clones were statistically significant.

#### 2.3.3. Quantification of MTXs by LC-MS/MS

None of the four clones produced MTX1, MTX2, or MTX4 in culture.

## 3. Discussion

The presence of selected, highly toxic species/strains (even if they may not be the numerically dominant ones) in *Gambierdiscus*/*Fukuyoa* blooms likely play a prominent role in CP outbreaks and severity [9,15,72]. Previous studies by Yogi et al. (2014) [57] and Ikehara et al. (2017) [2] also suggest that comprehensive data on the toxin profiles encountered in *Gambierdiscus* spp. blooms can provide baseline knowledge on the factors likely to influence fish ciguatoxicity in ciguatera-prone areas, and are therefore useful to inform CP risk models. The species *G. polynesiensis* is regarded as the dominant producer of CTXs in the food webs in French Polynesian lagoons [73]. The aim of the present study was to assess the overall toxin production and to characterize the toxin profiles in four strains of *G. polynesiensis* from distinct localities in French Polynesia, in relation to their growth stages.

In this study, two detection methods were used to evaluate the CTX production in *G. polynesiensis* strains, the functional assay CBA-N2a, and LC-MS/MS. Overall, good agreement was observed between the two methods, although slight quantitation variations were observed, e.g., LC-MS/MS detection method consistently yielded higher concentrations, in particular in the stationary growth phase. 

LC-MS/MS is designed to detect previously characterized CTX congeners and quantify those for which the corresponding reference standards are available. Here, absolute estimates of total concentrations are confounded by the lack of certified reference compounds (response factor assumption), and also the estimation made in expressing concentrations in P-CTX3C equivalents (TEF = 1 assumption). Still, the increase in total concentrations only observed in LC-MS/MS analyses is clearly due to the increased diversity of P-CTX3C analogs, for which the actual structure and toxicity of some analogs are currently unknown. For RIK7, the overall increase in toxin content is also due to the detection of P-CTX4A and –B above quantitation limit, two known analogs, to a lesser extent also due to P-CTX4A appearing in NHA4 and RG92. A further possible explanation for the differences noted between CBA-N2a and LC-MS/MS data may be that neuro-2a cytotoxicity will actually reflect the complex interactions on voltage-gated sodium channels (VGSCs) exerted by the combination of bioactive compounds present in *G. polynesiensis* culture extracts. In some instances, this may result in structural or chemical competition, as had been demonstrated for brevenal and brevetoxin in *K. brevis* [74]. 

Factors governing toxin biosynthesis in the genera *Gambierdiscus* and *Fukuyoa* are not well understood, and the role played by environmental and/or physiological drivers, genetic characteristics and symbiotic bacteria has been largely evoked [38,60]. In particular, the production of CTXs has been shown to significantly vary within and between species of *Gambierdiscu*s/*Fukuyoa*. A comprehensive review of the existing literature shows that toxic species in these genera can be classified into three distinct groups: (i) species with very low toxicity (of the order of fg P-CTX3C eq. cell^−1^), which comprises *G. belizeanus*, *G. caribaeus*, *G. carolinianus*, and *G. carpenteri*; (ii) species exhibiting a toxicity at sub-pg range (<1 pg P-CTX3C eq. cell^−1^), which includes *G. australes*, *G. pacificus*, *G. scabrosus*, *G. silvae*, *G. toxicus*, and (iii) a third group composed of *G. polynesiensis* and *G. excentricus* characterized by toxicities at pg range [54,75]. Thus, in the present study, CBA-N2a and LC-MS/MS data both confirmed the very high toxicity status of the species *G. polynesiensis* compared to other *Gambierdiscus* species, with all four clones exhibiting an overall toxin content ranging from 1.1 ± 0.1 up to 4.6 ± 0.7 pg P-CTX3C eq. cell^−1^, depending on the growth phase considered and the toxin detection method used. Based on the literature, only *G. excentricus* strains have shown comparable toxicity levels, i.e., 1.1 pg P-CTX1B eq. cell^−1^ [55], which would be equivalent to 3.5 pg P-CTX3C eq. cell^−1^ according to Bottein Dechraoui et al. (2007) [76]. These results are consistent with previously published data on the toxic potency of *G. polynesiensis* strains [24,47,49,56,77]. All these observations tend to reinforce the general belief that this species is likely the dominant source of CTXs entering the food web in the South Pacific region. Of note, Rhodes et al. (2014) [49] even reported a remarkably high CTX production in CAWD212, a Pacific strain of *G. polynesiensis* isolated from the Cook Islands (18.2 pg cell^−1^ as determined by LC-MS/MS). In contrast, so far, only one strain of *G. polynesiensis* isolated from Macauley Island (Kermadec Islands, New Zealand) has been found to produce neither MTX1 nor, unusually, any known CTXs [20].

Based on LC-MS/MS data, an increase in the production of CTXs and gambierone group analogs was observed from the exponential to the stationary phase in all four clones. Numerous studies have evaluated the effects of growth stage on the toxicity of *Gambierdiscus* spp. in vitro and have concluded that cellular toxin levels often increase with the age of the culture [24,43,67,68]. This enhanced toxicity noted at the stationary phase supports the common belief that CTXs are secondary metabolites, which are generally accumulated under deteriorating growth conditions such as those in batch cultures entering the stationary phase [68]. An increase in cellular toxin contents during the stationary phase has also been observed in several other dinoflagellates maintained in laboratory culture, e.g., azaspiracids in *Azadinium spinosum* [78] or okadaic acid and pectenotoxins in *Dinophysis* [79].

Another interesting result concerns the intraspecific variations clearly observed in the toxin production and/or toxin composition (i.e., identity and relative abundance of CTXs and gambierone group analogs) of *G. polynesiensis* strains from distinct geographic origins. Screening of toxicity is currently in progress on seven additional clones of *G. polynesiensis*, recently established in the laboratory, and will likely corroborate these preliminary observations. For example, NHA4 has been found to produce approximately four times more CTXs than TIO10 (1.20 ± 0.14 pg P-CTX3C eq. cell^−1^), a *G. polynesiensis* clone recently isolated from another locality in Nuku Hiva Island [56]. This strain-dependent production of CTXs is well established in *Gambierdiscus* and *Fukuyoa* spp. Indeed, toxic and nontoxic strains have been reported within the species *G. toxicus*, *G. australes*, and *G. pacificus* [15,24,41,44,45]. Likewise, Litaker et al. (2017) [54] found that the CTX production in *F. ruetzleri* originating from the Caribbean Sea was 27-fold and 3.8-fold more toxic than strains from the Gulf of Mexico and the Atlantic Ocean, respectively. In the same way, a comparison of the average toxicity of *F. paulensis* strains between three studies revealed that isolates from the Mediterranean Sea [22,40] were more potent than strains from the Pacific and Atlantic Oceans, in which no toxicity was detected [49]. However, in this latter case, some of these variations could merely result from differences in culture conditions, growth stage and/or toxin detection methods.

Clone RG92 was found significantly less toxic than RAI-1, RIK7, and NHA4. Of note, this clone clearly differed from the other three strains with regards to its age, as it has been held in the Institut Louis Malardé (Papeete, Tahiti, French Polynesia) culture collection since 1992. In contrast, RAI-1 was established in the laboratory in 2008, while clones RIK7 and NHA4 were isolated more recently. Interestingly, earlier studies reported an initial toxin content per cell of 4.3 pg P-CTX3C eq. cell^−1^ in RG92 [24], suggesting a progressive decline in the toxicity of this strain over time. A similar trend is presently observed in long-term cultures of strain TB92, another clonal isolate also established in the laboratory in 1992 [24], and whose composite cytotoxicity initially estimated at 5.8 ± 0.85 pg P-CTX3C eq. cell^−1^ [47] has gradually dropped to 3.6 pg P-CTX3C eq. cell^−1^ [77]. Likewise, Munday et al. (2017) [80] also mentioned significant changes in the CTX concentration of clone CAWD 212 from the Cook Islands [49]. All these findings suggest that high toxin production may not be a stable characteristic in long-term cultures of *G. polynesiensis*, contradicting earlier observations by Chinain et al. (2010) [24], however, more strains from diverse geographic origins should be examined to ascertain this hypothesis.

Another possible explanation for the decrease in RG92 average toxic potency noted here relates to its growth rate. The average growth rates exhibited by the four highly toxic clones of *G. polynesiensis* ranged from 0.22 ± 0.01 to 0.35 ± 0.09 day^−1^, and are consistent with those reported in the literature for other species of *Gambierdiscus* spp. when maintained under comparable temperature and salinity conditions [61,62,63,64,65]. Some studies, however, gave significantly lower growth rates such as those of Jang et al. (2018) [17] and Sparrow et al. (2017) [81] who reported growth rates not exceeding 0.07 and 0.14 day^−1^ in cultures of *G. jejuensis* and *G. carpenteri*, respectively. Of note, in their initial publication on the growth and toxin production of RG92 in culture, Chinain et al. (2010) [24] reported a rather low growth rate (i.e., 0.13 ± 0.03 day^−1^) and an average CTX production of 2.8 pg P-CTX3C eq. cell^−1^ for this strain, as determined by RBA. In the present study, RG92 exhibited a higher growth rate, i.e., 0.22 ± 0.02 day^−1^, which was associated with lower toxicity. The enhanced growth rate observed in this study likely results from the stabilization of pH in the cultures over the duration of the experiments. Several studies have examined the toxin production patterns in relation to growth rates in various *Gambierdiscus* species such as *G. polynesiensis*, *G. excentricus*, *G. australes*, and *G. carpenteri*, and concluded to a higher toxin biosynthesis in strains/species with low reproductive rates [24,44,54,67,82]. These findings suggest a trade-off between investments of cellular resources in growth vs. the production of defensive compounds [24,54], a pattern also described in another closely related harmful algal species, *Karenia brevis* [83,84]. In any case, these observations on the potential effects of physiological factors (e.g., cell division rate and age of culture) on *Gambierdiscus* spp. toxin production highlight the importance of considering both the culture conditions and harvest time if the pursued objective is to maximize toxin yields in cultures of toxigenic *Gambierdiscu*s spp. 

The study by Chinain et al. (2010) [24], the first to document the toxin profiles in two strains of *G. polynesiensis* (TB92 and RG92), initially reported the production of five of the major CTX congeners previously characterized in the ciguatera food chain, i.e., P-CTX3C, P-CTX3B, P-CTX4A, P-CTX4B, and M-seco-P-CTX3C. Thanks to the significant improvement of LC-MS/MS detection methodology, Sibat et al. 2018 [51] further detected four additional CTX derivatives of the P-CTX3C group in cultures of TB92, including three analogs whose structural identification is still pending. Here, the LC-MS/MS analysis of four additional clones of *G. polynesiensis* from distinct geographic origins yielded similar results, suggesting that the toxin profile is a stable characteristic in this species as previously proposed by Chinain et al. (2010) [24]. The occurrence of several oxidized toxins in all four clones of *G. polynesiensis*, e.g., 2-OH-P-CTX3C and M-seco-P-CTX3C, substantiate earlier findings by Yasumoto et al. (2000; 2001) [25,59], Yogi et al. (2011) [26], and Laza-Martinez et al. (2016) [40] who also reported oxidized forms in culture extracts of *G. toxicus* and *F. paulensis*, respectively. Rather unusual, Roeder et al. (2010) [50] also reported the occurrence of another oxidized form (i.e., 2,3-dihydroxy-P-CTX3C) in several *Gambierdiscus* strains, while no CTX3B or -3C were present in these strains. However, according to Pisapia et al. [29], this compound may have been misidentified with 44-methylgambierone. In any case, these observations highlight the need to re-examine the hypothesis that CTXs are mainly oxidized in fish during food chain transmission [26].

In this study, gambierone, a compound first described by Rodriguez et al. (2015) [36] in *G. belizeanus*, was demonstrated for the first time in cultures of *G. polynesiensis*. The detection of 44-methylgambierone in all four clones also confirmed the ubiquitous character of this compound that seems to be produced by all *Gambierdiscus* and *Fukuyoa* species described to date, except for *G. excentricus* [29], as well as *F. yasumotoi* and *G. jejuensis* for which data are not yet available [38]. Conversely, none of the four clones produced MTX1, 2 or -4, a result also consistent with those of Rhodes et al. (2014) [49] for *G. polynesiensis* CAWD212 originating from the Cook Islands. These observations suggest that the common belief that almost all *Gambierdiscus* isolates produce MTXs while many fail to produce detectable levels of CTXs needs to be further evaluated. We postulate that the production of MTX analogs is likely to be species-specific within *Gambierdiscus* and *Fukuyoa* species as well [29].

The production pattern described in many *Gambierdiscus* species, including *G. polynesiensis*, clearly suggests a trade-off between reduced growth and increased toxin synthesis and poses the question of the ecological relevance of the toxins synthesized by *Gambierdiscus*/*Fukuyoa* spp. The diversity and structural complexity of ciguatera toxins have led to the speculation that allelochemical agents may be used by *Gambierdiscus* to compete with other epiphytic dinoflagellates for space or limit grazing by herbivores [3,24,54,69,85,86]. This hypothesis is strongly supported by the results of several studies that have examined the interactions between *Gambierdiscus* spp. and co-occurring taxa of the thallisphere: for instance, *Prorocentrum concavum* and *G. toxicus* were reciprocally inhibited both in exudate-supplemented cultures and in cross-cultures [87,88]. Moreover, both exudates and CTX extracts of *G. toxicus* were able to inhibit the growth of several diatoms and chlorophytes species [64]. The study by Sugg and Van Dolah (1999) [87] also showed that, in crossed culturing experiments, culture media preconditioned with filtered-exudates of *G. toxicus* were able to inhibit the growth of *Coolia monotis* and *Ostreopsis lenticularis*. However, the exact nature of the compounds involved in these complex mechanisms remains to be elucidated. Of note, one study has highlighted the capability of gambieric acids to act as an endogenous growth enhancer toward *G. toxicus* cultures [89]. Interestingly, these compounds, which are excreted in the culture medium by *G. toxicus* at the late growth phase [90], are also known to exhibit potent antifungal activities against filamentous fungi.

Gambierone and 44-methylgambierone, which are found in high concentrations in both cells and culture medium of *G. polynesiensis* in the present study, are also known for their bioactive properties on cell membranes: e.g., gambierones seem to share similar biological activities with CTXs, in that they cause VGSC activation in a similar pattern as CTXs, although in much lower intensity [36]. In addition, both gambierone and 44-methylgambierone induce a small rise in the cytosolic calcium concentration in human cortical neurons, like CTXs [34], while the expression of ionotropic glutamate receptor has been shown to be modified by chronic exposure of human neurons to gambierones, an alteration that could be involved in the neurological manifestations observed in human CP [34]. Likewise, the ability of gambieric acids to bind to the site 5 of VGSCs, although with lower affinity than CTXs, has been previously described by Inoue et al. (2003) [91].

Taken together, all these observations strongly suggest that, besides CTXs, many of the other metabolites produced by *G. polynesiensis* may be implicated in CP although their structure and effects on cellular membranes seem to differ from those typical of CTXs. The data presented here also clarify the relationship between the diversity of toxic compounds produced by *G. polynesiensis* and the exceptionally high toxic activity of this species.

## 4. Conclusions

This study examined how toxicity and toxin profiles vary with the growth phase and by geographic origin in four clones of *G. polynesiensis*, the primary source of ciguatoxins in the South Pacific region. Results confirmed the very high toxicity of this species and the high diversity of toxic metabolites produced by *G. polynesiensis* in culture. Besides, an array of CTX analogs (at least nine distinct CTX derivatives), the presence of noticeable amounts of gambierone, and 44-methylgambierone were also observed in cell extracts, whereas no MTX1 or -4 were detected. Results also highlighted significant intraspecific variations in toxin production between these clones, although the relationship between the diversity of toxic compounds produced and regional differences in CP risk remains to be clarified. Confirmation that *G. polynesiensis* toxicity significantly varies with culture age suggests the necessity for further studies on the impact of various stressors such as temperature, pH, availability, and/or source of nitrogen on the in vitro toxin production of *G. polynesiensis*. In fine, such studies will also contribute to decrypting the link between global change (e.g., ocean warming and acidification, eutrophication) and increased CP risk worldwide. Furthermore, they will greatly benefit current efforts aimed at obtaining suitable quantities of certified analytical standards for CP research. Concurrently, in-depth investigations on the compatibility between several detection methods such as LC-MS/MS, CBA-N2a, and RBA that are classically used in CP risk monitoring programs, will provide a better understanding of quantitation differences across methods, and foster trans-regional comparative studies on ciguatera.

## 5. Materials and Methods 

### 5.1. Source of G. polynesiensis Isolates

The four clones of *G. polynesiensis* used in this study (RIK7, NHA4, RAI-1, and RG92, originate from different islands of French Polynesia with contrasted CP risk [92,93,94]: i.e., Mangareva Island (Gambier archipelago), Nuku Hiva (Marquesas archipelago), Raivavae (Australes archipelago), and Rangiroa (Tuamotu archipelago) (Table 6). These clones were isolated from macro-algal and/or artificial substrate (window screen) samples collected following the methods described in Chinain et al. (2010) [24] and Tester et al. (2014) [95], respectively. All four strains are part of the Laboratory of Marine Biotoxins culture collection at the Institut Louis Malardé (Tahiti, French Polynesia), where cultures are deposited.

### 5.2. In Vitro Cultures

All four *G. polynesiensis* clones selected for this study were acclimated to the culture conditions described hereafter for years prior to experimentation (Table 6). For each clonal isolate, batch cultures of *G. polynesiensis* clones were established in triplicate by inoculating 200 mL of natural seawater in 250 mL Erlenmeyer flasks. Seawater was previously aged for 1 month, filtered through a 0.2 μm GF/F Whatman filters (Dutscher, Brumath, France), and sterilized by autoclaving for 20 min at 120 °C. To optimize cell growth, seawater was supplemented with 200 μL of “f_10k_” culture medium (Holmes et al., 1991) [66]. Cultures were grown in an incubator (ThermoStable IR250, Wigel, Eichel, Germany) at 26 ± 1 °C, under 60 ± 10 μmol photons m^−2^s^−1^ irradiance, in a 12 h light: 12 h dark photoperiod. Flasks were incubated at randomly determined sites in the incubator (flask position was changed on a daily basis). The pH of the culture medium was stabilized at 8.4 by addition of a few drops of 1 M HCl once or twice a day. 

### 5.3. Growth Rate 

Preliminary assays confirmed a linear correlation between in vivo fluorescence (relative fluorescence units or RFU) measured using a spectrophotometer (Victor×2, PerkinElmer, Wellesley, Massachusetts, MA., USA) and *G. polynesiensis* cell densities assessed using a Multisizer III^TM^ particle counter fitted with a 280 mm aperture tube and calibrated using protocols provided by the manufacturer (Beckman Coulter, Inc., Brea, CA., USA). Standard linear regression lines (RFU vs. cell densities) gave R^2^ values between 0.95 and 0.98, indicating in vivo fluorescence was a good proxy for cell abundance (Figure 6).

Thereby, in all further experiments, cell growth was assessed by measuring culture fluorescence at 485 nm, at two days intervals. To limit error during fluorescence measurements, cultures were mixed fully prior to fluorescence reading then a 100 µL aliquot of each culture was deposited in 96-well plates where cells were lysed by the addition of 100 µL di-methyl sulfoxide (DMSO; n = 3 wells per batch culture). Five fluorescence readings were performed on each well. 

Growth rates (µ) were determined using the following formula:µ = [ln × (rfu_t1_/rfu_t0_)]/[ln (2) × (t_1_–t_0_)]
in which μ (division day^−1^) is the growth rate, rfu_t1_ and rfu_t0_ represent the fluorescence measured at times t_1_ (days) and t_0_, respectively, corresponding to the exponential growth phase portion of the growth curve. 

### 5.4. Cells Harvest and Toxin Extraction

Prior to cell harvest, the total cell yield was determined by automated counting on the Multisizer III^TM^ particle counter. Cultures were harvested by filtration onto 90 mm 0.2 µm GF/F Whatman filters (Dutscher, France) in their late exponential and early stationary growth phase (i.e., at t_8_ and t_18_ post-inoculation, respectively). Each filter was then transferred into a 50 mL Greiner tube, and freeze-dried for 20 h at −20 °C, 1 mbar, then 4 h at −60 °C, 0.01 mbar (Martin Christ, Beta 1-8 LDplus). Each filter-bearing cell sample was then stored at 4 °C until further extraction.

*G. polynesiensis* cells were extracted by adding 20 mL of methanol (MeOH) directly into the tubes containing each freeze-dried cell sample. Tubes were then vortexed for a few seconds and then placed in an ultrasound tank for one hour. Following a centrifugation step at 2800 *g* for 10 min, the resulting supernatant was recovered in a 250 mL flask. This extraction step was repeated once in MeOH and then twice in aqueous methanol (MeOH:H_2_O 1:1). The four supernatants were further pooled and dried under vacuum using a rotary evaporator (Rotavapor RII, Büchi, Roubaix, France), and the resulting crude cellular extract resuspended in 6 mL of pure methanol. 

Half of this solution (3 mL MeOH) was filtered through Sartorius filter (3 cm diameter, 0.45 μm porosity), then evaporated under nitrogen to dryness and reconstituted in 1 mL MeOH to obtain a crude extract destined for the screening of both MTXs and gambierone group toxins by LC-MS/MS. The remaining 3 mL were further partitioned between 50 mL of dichloromethane (CH_2_Cl_2_) and 2 × 25 mL of aqueous MeOH (MeOH:H_2_O 3:2) according to the procedure previously described in Chinain et al., 2010b, to separate lipid-soluble compounds (e.g., CTXs) from water-soluble compounds. The resulting dichloromethane-soluble fractions (DSFs) were dried under vacuum, weighed and stored at 4 °C until tested for their toxicity using the Neuroblastoma Cell-Based Assay (CBA-N2a).

In addition, following cell harvest, the potential presence of toxic compounds in the culture media was also investigated using the protocol described in Roué et al. (2018) [70]. To this end, 2 g of HP-20 resin were exposed to each culture medium (200 mL per clonal culture) during 72 h, at room temperature, and under permanent stirring. Following filtration, the resin was thoroughly washed with milliQ water and the toxins extracted with 50 mL pure methanol. The resulting crude extract was dried under vacuum then stored at 4 °C until tested by LC–MS/MS.

### 5.5. Neuroblastoma Cell-Based Assay (CBA-N2a)

The CBA-N2a analyses were conducted following the procedure previously described in Darius et al. (2018) [56], with minor modifications. Briefly, a density of 45,000 neuroblastoma (neuro-2a) cells/200 µL/well were seeded in 5% fetal bovine serum RPMI-1640 supplemented medium, in 96-well microtiter plates further kept at 37 °C in a humidified 5% CO_2_ atmosphere. After 24 h of growth (when the cell layer has reached 100% confluence), the culture medium was replaced by 200 µL of 2% fetal bovine serum RPMI-1640 supplemented medium for half of wells, and the same medium containing an ouabain-veratridine solution (OV) at a final concentration of 76.2/7.62 µM for the other half of the wells. 

Untreated cells (i.e., without the addition of OV solution, OV^−^ conditions) or treated cells (i.e., with the addition of OV solution, OV^+^ conditions) were first exposed to a serial dilution 1:2 of eight concentrations of P-CTX3C ranging from 0.15 to 19.05 fg µL^−1^. The maximum concentration of dry extract (MCE) that does not induce unspecific mortalities in neuro-2a cells was established at 9500 pg/µL [56]. Thereby, all lipid-soluble fractions (i.e., DSFs) were tested in CBA-N2a using a serial dilution 1:2 of eight concentrations ranging from 25.1 to 0.2 pg µL^−1^ of dry extract, in order to generate a full sigmoidal dose-response curve. Each concentration was tested in OV^−^ and OV^+^ conditions, in triplicate per plate. Following an overnight incubation period, viability of neuro-2a cells was assessed using the MTT assay [56]. The absorbance was measured at 570 nm using a plate reader (iMark Microplate Absorbance Reader, BioRad, Marnes la Coquette, France). For all experiments, absorbance values of OV^−^ and OV^+^ control wells were around 1.2, corresponding to 100% viability. Absorbance data were fitted to a sigmoidal dose–response curve (variable slope) based on the four-parameter logistic model (4PL) allowing the calculation of EC_50_ values (i.e., the tested cell concentration inducing a loss of 50% of the higher absorbance value) in cells eq. µL^−1^ of dry extract using Prism v8.1.2 software (GraphPad, San Diego, CA, USA). The cell toxin content (Tx) for each tested clone was further estimated using the following formula: Tx (in pg P-CTX3C eq. cell^−1^) = (EC_50_ of P-CTX3C/ EC_50_ of sample)

The limit of detection (LOD = (EC_80_ of P-CTX3C/MCE *Gambierdiscus*)) and the limit of quantification (LOQ = (EC_50_ of P-CTX3C/MCE *Gambierdiscus*)) of CBA-N2a for *Gambierdiscus* spp. were estimated at 0.17 fg P-CTX3C eq. cell^−1^ and 0.34 fg P-CTX3C eq. cell^−1^, respectively, according to Roué et al. (2016) [47].

### 5.6. Liquid Chromatography Coupled with Tandem Mass Spectrometry (LC-MS/MS) Analysis

In this study, two LC-MS/MS acquisition methods were used to determine the toxin profile of the strains of *G. polynesiensis*: method 1 for the P-CTX group based on the protocol by Sibat et al., 2018 [51] and method 2 for MTXs and gambierone group toxins [29]. A brief description is given below.

Both experiments were performed using a UHPLC system (UFLC Nexera, SHIMADZU, Kyoto, Japan) coupled to a hybrid triple quadrupole-linear ion-trap API4000 QTRAP mass spectrometer (SCIEX, Redwood city, CA, USA) equipped with a TurboV^®^ electrospray ionization source (ESI). The instrument control, data processing, and analysis were conducted using Analyst software 1.6.2 (Sciex, CA, USA).

#### 5.6.1. Method 1: P-CTXs Group 

A linear gradient using water as eluent A and MeOH as eluent B, both eluents containing 2 mM ammonium formate and 50 mM formic acid, was run on a Zorbax Eclipse Plus C18 column, 50 × 2.1 mm, 1.8 μm, 95 Å (Agilent Technologies, Santa Clara, CA, USA). The flow rate was 0.4 mL min^−1^, the injection volume was 5 μL, and the column temperature 40 °C. 

The elution gradient was as follows: 78% B to 88% B from 0 to 10 min, hold at 88% B for 4 min, decrease from 88% to 78% in 1 min and hold for 5 min at 78% B. Mass spectrometric detection was performed in positive ionization mode using Multiple Reaction Monitoring (MRM) scanning. 

The optimized ESI+ parameters were set as follows: curtain gas at 25 psi, ion spray at 5500 V, turbo gas temperature at 300 °C, gas 1 and 2 were set at 40 and 60 psi, respectively, and an entrance potential at 10 V. 

MRM acquisition method was created using the scheduled MRM algorithm. This algorithm optimizes dwell times and cycle time to provide better peak detection and improve reproducibility. A detection window of 90 sec and a target scan time of 2 sec were chosen for the MRM method. Calibration solution of P-CTX3C (Wako, Tokyo, Japan) was prepared in MeOH, with concentration ranging from 12 to 200 ng mL^−1^.

#### 5.6.2. Method 2: MTXs and Gambierone Group

A linear gradient using water as eluent A and 95% acetonitrile as eluent B, both eluents containing 2 mM ammonium formate and 50 mM formic acid, was run on a Kinetex C18 column, 50 × 2.1 mm, 2.6 μm, 100 Å (Phenomenex, Torrance, CA, USA). The flow rate was 0.4 mL min^−1^, the injection volume was 5 μL, and the column temperature 40 °C.

The elution gradient was as follows: 10% B to 95% B from 0 to 10 min, hold at 95% B for 2 min, decrease from 95% to 10% in 1 min, and hold for 3 min to equilibrate. Mass spectrometric detection was performed in negative ionization mode using MRM scanning. The *m*/*z* transitions used were listed in Table 7.

The optimized ESI^-^ parameters were set as follows: curtain gas at 25 psi, ion spray at −4500 V, turbo gas temperature at 500 °C, gas 1 and 2 at 50 psi, declustering potential at −210 V, and an entrance potential at −10 V. 

To quantify the MTXs and Gambierone toxins, a calibration solution of MTX1 (Wako, Japan) was prepared in 50% aq. MeOH with concentration ranging from 0.2 to 5.0 µg mL^−1^.

### 5.7. Statistical Analyses

To determine whether growth rates and toxicity differences observed among clones were statistically significant, standard deviations (SD) were calculated, then depending on normality and homoscedasticity of data, an analysis of variance (ANOVA) (normality, homoscedasticity) or a Kruskal-Wallis and Wilcoxon test (non-normality and homoscedasticity) were performed using RStudio v1.0.153 (RStudio, Inc., Boston, MA, USA). Confidence intervals were calculated at 95% (*p*-value < 0.05). 

## Figures and Tables

**Figure 1 toxins-11-00735-f001:**
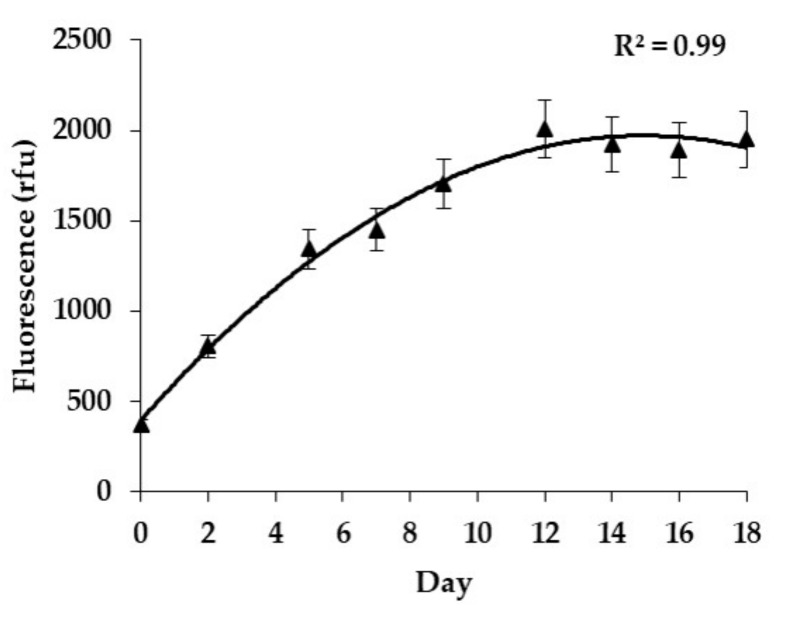
Growth curve of *Gambierdiscus polynesiensis* RAI-1 as measured by mean natural in vivo fluorescence (rfu). Data represent the mean ± SD of three experiments (with each fluorescence reading run five times).

**Figure 2 toxins-11-00735-f002:**
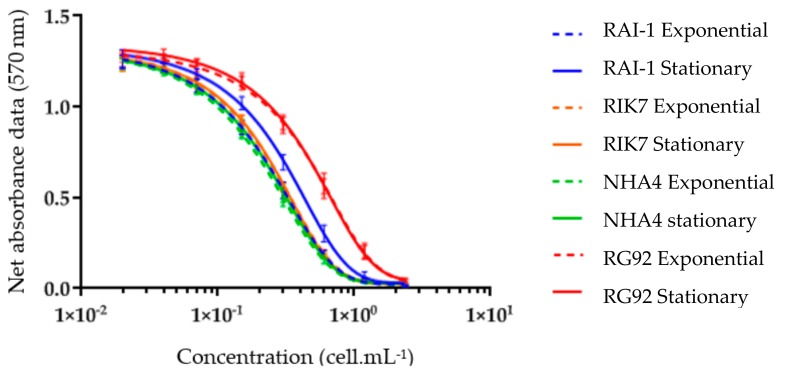
Dose-response curve of Neuro-2a cells in OV^+^ conditions when exposed to increasing concentrations of dichloromethane-soluble fractions of *G. polynesiensis* strains. Data represent the mean ± SD of three replicate cultures per clonal isolate, with each concentration run in triplicate.

**Figure 3 toxins-11-00735-f003:**
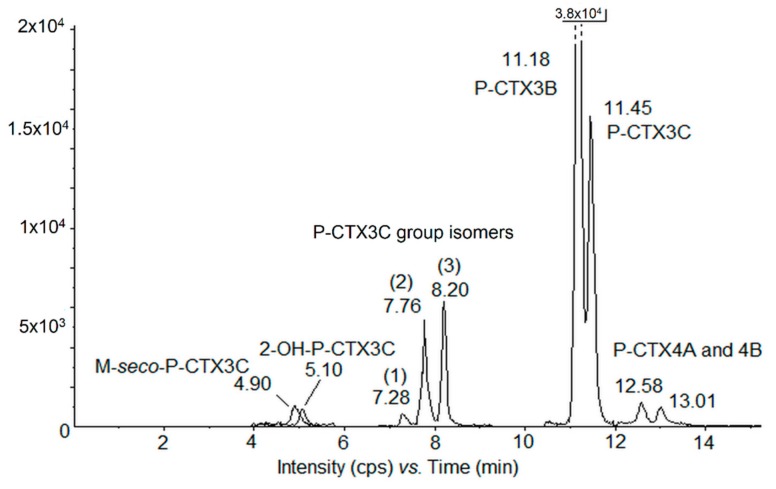
Chromatogram of scheduled Multiple Reaction Monitoring (MRM) liquid chromatography–tandem mass spectrometry (LC-MS/MS) showing the different P-CTX analogs detected in crude extracts (methanol) of strain NHA4 tested at the stationary growth phase. The retention time (RT) of each analog is indicated close to the corresponding peak and the different P-CTX3B/C group isomers 1, 2, and 3 are indicated in brackets. NB: P-CTX3B peak is cut off to show all analogs.

**Figure 4 toxins-11-00735-f004:**
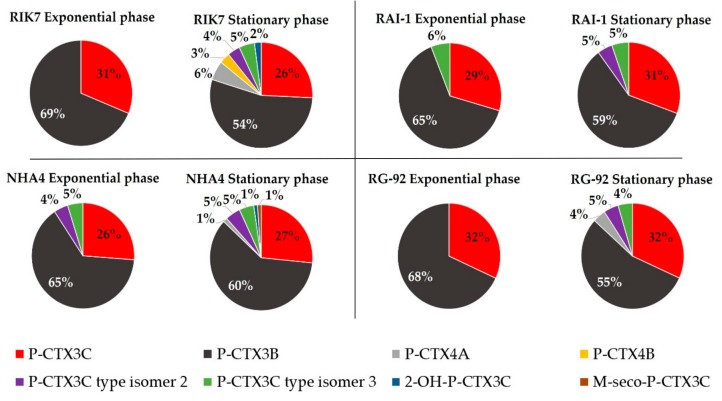
Comparison of the toxin profiles in batch cultures of *G. polynesiensis* RIK7, NHA4, RAI-1, and RG92 as determined by LC-MS/MS at the exponential vs. the stationary phase. The relative abundance of each analog is expressed in percentage relative to the total toxin content. Only quantifiable compounds are included in this representation.

**Figure 5 toxins-11-00735-f005:**
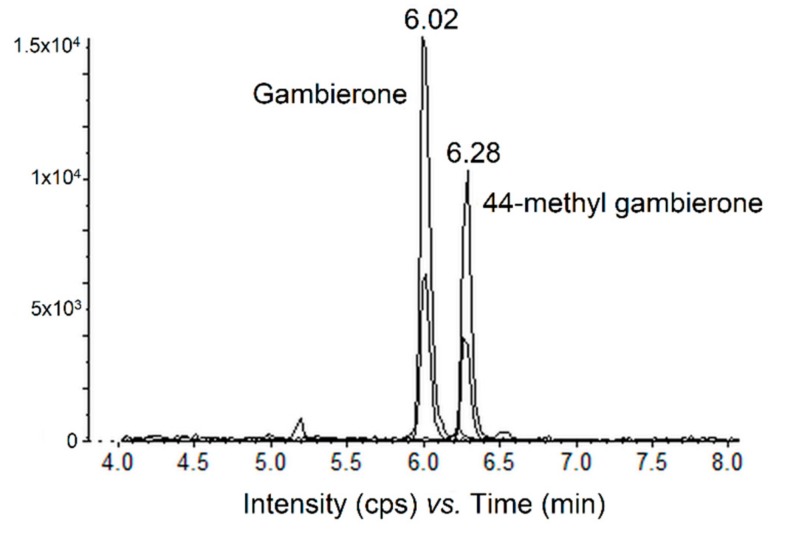
Chromatogram of non-scheduled MRM LC-MS/MS (negative mode) showing the presence of gambierone and 44-methylgambierone in crude extracts (methanol) of strain NHA4 at the stationary growth phase.

**Figure 6 toxins-11-00735-f006:**
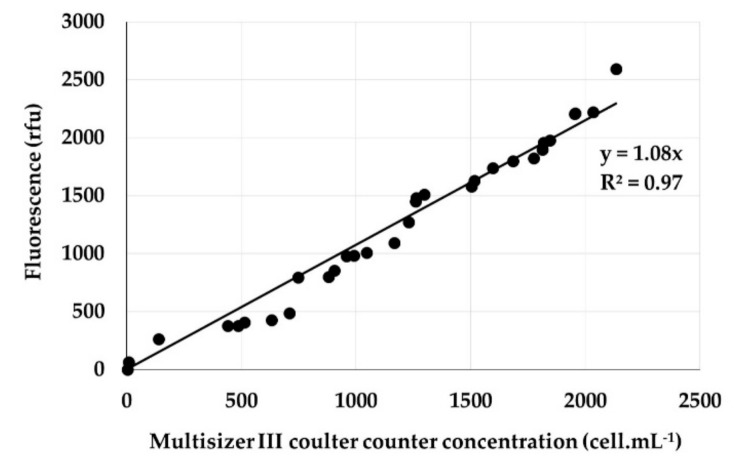
Linear relationship between *G. polynesiensis* RAI-1 cell concentration (cells.mL^−1^) and in vivo fluorescence (rfu, relative fluorescence units). Filled circles represent individual measurements to which a linear regression line has been fitted.

**Table 1 toxins-11-00735-t001:** Growth rate for each *G. polynesiensis* clone, as determined at the exponential phase.

Parameter	RIK7	NHA4	RAI-1	RG92
Growth rate µ *	0.22 ± 0.01	0.26 ± 0.03	0.35 ± 0.09	0.22 ± 0.02

* Growth rates µ are expressed in div. day^−1^ ± SD (n = 3).

**Table 2 toxins-11-00735-t002:** Composite toxicity of *G. polynesiensis* clones as measured by neuroblastoma cell-based assay (CBA-N2a) at the exponential vs. stationary phase.

CLONE	Growth Phase	EC_50_ (Cell. mL^−1^) ^(1,2)^	pg P-CTX3C eq. Cell^−1 (2)^
RIK7	Exponential	0.265 ± 0.01	3.3 ± 0.1
	Stationary	0.266 ± 0.02	3.3 ± 0.2
NHA4	Exponential	0.234 ± 0.02	3.7 ± 0.3
	Stationary	0.248 ± 0.03	3.6 ± 0.4
RAI-1	Exponential	0.253 ± 0.02	3.5 ± 0.3
	Stationary	0.334 ± 0.05	2.6 ± 0.4
RG92	Exponential	0.552 ± 0.09	1.6 ± 0.2
	Stationary	0.528 ± 0.05	1.7 ± 0.2

^1^ EC_50_ is the effective concentration reducing 50% of cell viability. ^2^ Effective concentration (EC_50_) and composite cytotoxicity data represent the mean ± SD of three replicate cultures per clonal isolate.

**Table 3 toxins-11-00735-t003:** LC-MS/MS quantitation (pg P-CTX3C eq. cell^−1^) of P-CTXs analogs in *G. polynesiensis* RIK7, NHA4, RAI-1, and RG92, at the exponential (Expo.) vs. stationary (Stat.) growth phase, as approached by integration of the chromatogram peaks. Data represent the mean ± SD of three replicate cultures per clonal isolate, with a single LC-MS/MS run per extract). The limit of quantification (LOQ) was established at 40 ng P-CTX3C mL^−1^ (this ranges from 0.02 to 0.08 pg equivalents P-CTX3C cell^−1^). Total concentrations are the sum of individual concentrations which were estimated in P-CTX3C equivalents, assuming equal response factors in LC-MS/MS.

CLONE/Growth Phase	P-CTX3C	P-CTX3B	P-CTX4A	P-CTX4B	P-CTX3B/C Type Isomer 2	P-CTX3B/C Type Isomer 3	2-OH-P-CTX3C	M-seco-P-CTX3C	Total
RIK7 Expo.	1.00 ± 0.14	2.21 ± 0.37	<LOQ	<LOQ	<LOQ	<LOQ	<LOQ	<LOQ	3.21 ± 0.51
RIK7 Stat.	1.08 ± 0.05	2.27 ± 0.07	0.25 ± 0.04	0.14 ± 0.03	0.16 ± 0.01	0.21 ± 0.01	0.09 ± 0.01	<LOQ	4.20 ± 0.22
NHA4 Expo.	1.06 ± 0.03	2.61 ± 0.19	<LOQ	<LOQ	0.18 ± 0.01	0.20 ± 0.01	<LOQ	<LOQ	4.05 ± 0.24
NHA4 Stat.	1.23 ± 0.18	2.79 ± 0.45	0.06 ± 0.01	<LOQ	0.22 ± 0.02	0.21 ± 0.03	0.05 ± 0.01	0.05 ± 0.01	4.61 ± 0.71
RAI-1 Expo.	0.96 ± 0.24	2.10 ± 0.52	<LOQ	<LOQ	<LOQ	0.20 ± 0.04	<LOQ	<LOQ	3.26 ± 0.8
RAI-1 Stat.	1.11 ± 0.02	2.15 ± 0.03	<LOQ	<LOQ	0.17 ± 0.01	0.19 ± 0.01	<LOQ	<LOQ	3.62 ± 0.07
RG92 Expo.	0.35 ± 0.04	0.75 ± 0.05	<LOQ	<LOQ	<LOQ	<LOQ	<LOQ	<LOQ	1.10 ± 0.09
RG92 Stat.	0.71 ± 0.01	1.23 ± 0.11	0.09 ± 0.02	<LOQ	0.10 ± 0.02	0.10 ± 0.02	<LOQ	<LOQ	2.23 ± 0.18

**Table 4 toxins-11-00735-t004:** Results of the ANOVA test relative to the toxicity differences observed between clones and growth phases. A “*p*-value” < 0.05 indicates statistically significant differences in total toxin concentration (LC-MS/MS).

Paired Comparison	Difference	*p* Value
RG92–RIK7	−2.18	0.0000004
RG92–NHA4	−2.65	0.0000000
RG92–RAI-1	−1.75	0.000005
RIK7–NHA4	−0.46	0.173
RIK7–RAI-1	0.44	0.215
NHA4–RAI-1	0.90	0.004
Stationary-Exponential	0.71	0.0003

**Table 5 toxins-11-00735-t005:** Sum of intra-plus extracellular concentrations (Σ_intra + extra_) of 44-methylgambierone and gambierone (in pg eq. MTX1 cell^−1^) as well as distribution (%) between cells (Cell.) and culture medium (Med.) of *G. polynesiensis* strains, at the exponential (Expo.) vs. stationary (Stat.) phases. The LOQ was established at 40 ng P-CTX3C mL^−1^. Only one batch culture was analyzed per clonal isolate in a single run. Hence, it was not possible to test for the statistical significance of variations between strains.

Analyte/Sum	RIK7 Expo.		RIK7 Stat.		NHA4 Expo.		NHA4 Stat.		RAI-1 Expo.		RAI-1 Stat.		RG92 Expo.		RG92 Stat.	
Σ_intra + extra_ Gambierone (pg eq. MTX1 cell^−1^)	256.5		329.1		58		114.1		135.4		128.2		97.5		235.3	
Σ_intra + extra_ 44-methylgambierone (pg eq. MTX1. cell^−1^)	118.5		139.7		25		35.4		20.8		17		19.3		41.5	
Total (pg eq. MTX1 cell^−1^)	375		468.8		83		149.5		156.2		145.2		116.8		276.8	
	Cell.	Med.	Cell.	Med.	Cell.	Med.	Cell.	Med.	Cell.	Med.	Cell.	Med.	Cell.	Med.	Cell.	Med.
Gambierone (%)	53	47	70	30	79	21	33	67	98	2	90	10	99	1	89	11
44-methylgambierone (%)	26	74	53	47	46	54	16	84	100	-	68	32	100	-	73	27

**Table 6 toxins-11-00735-t006:** Geographic origin and year of isolation of the four *G. polynesiensis* clones selected for this study. Incidence rates (I.R., number of cases/10,000 population) are derived from the number of CP cases reported in 2016 and are used as a proxy of CP risk.

Year/Origin/Incidence	RIK7	NHA4	RAI-1	RG92
Year of isolation Archipelago	2013 Gambier	2015 Marquesas	2008 Australes	1992 Tuamotu
Island	Mangareva	Nuku Hiva	Raivavae	Rangiroa
I.R.	354	51	43	N/A ^1^

^1^ N/A: data not available. Source of data: Institut Louis Malardé (www.ciguatera.pf).

**Table 7 toxins-11-00735-t007:** List of MRM transitions (*m*/*z*) used for method 2 on API4000 QTrap.

Compound	MRM Transitions (*m*/*z*)	
MTX1	1689.8 > 1689.6	[M-2H]^2−^/[M-2H]^2−^
	1689.8 > 96.9	[M-2H]^2−^/[HOSO3]^2−^
	1126.2 > 1126.2	[M-3H]^3−^/[M-3H]^3−^
	1126.2 > 96.9	[M-3H]^3−^/[HOSO3]^3−^
MTX2	1637.5 > 1637.5	[M-2H]^2−^/[M-2H]^2−^
	1637.5 > 96.9	[M-2H]^2−^/[HOSO3]^2−^
	1091.5 > 1091.5	[M-3H]^3−^/[M-3H]^3−^
	1091.5 > 96.9	[M-3H]^3−^/[HOSO3]^3−^
MTX4	1646.2 > 1646.2	[M-2H]^2−^/[M-2H]^2−^
	1646.2 > 96.9	[M-2H]^2−^/[HOSO3]^2−^
Gambierone	1023.5 > 1023.5	[M-H]^−^/[M-H] ^−^
	1023.5 > 96.9	[M-H]^−^/[HOSO3]^−^
44-methylgambierone	1037.6 > 1037.6	[M-H]^−^/[M-H]^−^
	1037.6 > 96.9	[M-H]^−^/[HOSO3]^−^

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
