# Peer review of "Intraspecific Variability in the Toxin Production and Toxin Profiles of In Vitro Cultures of Gambierdiscus polynesiensis (Dinophyceae) from French Polynesia"

_toxins, 2019, doi:10.3390/toxins11120735_

Round 1

Reviewer 1 Report

The paper reports the evaluation of the ciguatoxin production in four clones at different growth stage from distinct locations in French Polynesia. The assessment of the toxin content was made by two alternative methods. The LC-MS/MS analysis allowed to determine the toxin profile. The study disclosed some interesting, unreported results, such as the presence of three undescribed CTX analogues, the presence of gambierone and its 44-methyl analogue, the absence of MTXs and a high variation of the chemical diversity of produced toxins in relation to growth phase.The analysis is high professional in quality and the interesting results were clearly discussed.

Author Response

Thank you for your kind comment.

English language has been improved.

Reviewer 2 Report

The major problem of the manuscript is the lack of novelty and the absence of novel ideas brings easily predictable results. Therefore this manuscript will attract very limited interest. Evidently, that different strains of dinoflagellates could produce different amounts of metabolites and it could be dependent on growth phase and conditions. Moreover, this scientific field was mostly covered by other studies ( of this group, I suppose). Nevertheless, of course, it was important to establish an increase in toxin production from the exponential to the stationary phase.

The manuscript quite well written but needs additional proofreading. All methods were adequate, and results deserve to be believed. 

If the Editor of the special issue considers this manuscript suitable than it could be published with minor revision. 

Minor issues 

A lot of awkward sentences, please, try to rephrase

 for example (110-112 )

Furthermore,

 111 CTX-related toxicity data as determined by CBA-N2a and LC-MS/MS analysis will also be compared 

112 in order to evaluate the good agreement between these two methods.

Author Response

Line 109: the sentence “Furthermore, CTX-related toxicity data as determined by CBA-N2a and LC-MS/MS analysis will also be compared in order to evaluate the good agreement between these two methods” has been modified into “Furthermore, CTX-related toxicity, as determined by CBA-N2a, and concentrations of CTX-analogs, as determined by LC-MS/MS analysis, will also be compared to each other to evaluate whether or not there is an agreement between these two methods”.

We have also entirely revised the manuscript for English language, in particular the length of sentences and potentially awkward expressions.